# The Relationships among Sport Participation Level, Flow Experience, Perceived Health Status and Depression Level of College Students

**DOI:** 10.3390/ijerph20010251

**Published:** 2022-12-23

**Authors:** Suh-Ting Lin, Ying-Hua Hung, Meng-Hua Yang

**Affiliations:** 1Office of Physical Education, Tamkang University, New Taipei City 251301, Taiwan; 2Office of Physical Education, National Formosa University, Yunlin 632301, Taiwan; 3Department of Physical Education, Health and Recreation, National Chiayi University, Chiayi 621302, Taiwan

**Keywords:** sport participation level, flow experience, self-rated health, depression level

## Abstract

The purpose of this study was to examine the relationships between sport participation level, flow, perceived health status and depression using gender and grades as control variables of college students in Taiwan. Based on previous research, the study established the proposed model: using sport participation level and flow experience as predicting variables, perceived health status and depression as dependent variables, and gender and grades as control variables. A total of 700 structured questionnaires were distributed to college students using convenience sampling among seven universities in Taiwan with a valid return rate of 86.5%. Structural equation modeling was used to test the relationships among the above-mentioned variables. The study found: 1. Male students had higher self-rated health perception than female students. 2. Students with higher grades perceived higher levels of depression than those with lower grades. 3. Among all variables, the level of sport participation had a positive predicting power of perceived health status and a negative predicting power of depression level; perceived health status had a negative predicting power of depression; while flow had no moderating effect among sport participation level, perceived health status and depression. In the model, the predicting variables had a predicting power of 0.58 (R^2^) for depression, indicating a good model. Conclusions and implications were made according to the findings of the study.

## 1. Introduction

According to World Health Organization [1], one-sixth of the population consists of adolescents aged from 10–19 years old. Among this group, depression ranked as the fourth factor attributed to disabilities and diseases, which made them isolated, feel lonely, and even worse, resulted in suicide. According to WHO, suicide was ranked as the third cause of death in 2016. According to the Taiwan Association Against Depression [2], there were over two million (8.9% of the population in Taiwan) suffering from middle levels of depression, and among them, nearly 1.25 million were diagnosed with high levels of depression. In addition, people suffering from depression are highly likely to commit suicide and end up dead. According to TADD’s statistics, suicide ranked as the second cause of death among 15–24-year-old youths and adults, and the suicide death toll increased from 193 in 2018 to 257 in 2019. Suicide cases increased from 4950 in 2019 to 7991 in 2020. Another report from the John Tung Foundation [3] investigated 5515 college students from 58 universities in Taiwan, and reports indicated 18.7% of them suffered from depression and needed further professional assistance. The above-mentioned statistics indicated one out of five college students suffered from depression and the situation has gotten even worse now. At present, there are 1.3 million registered college students in Taiwan, which indicates nearly 125 thousand students suffered from mild-to-high levels of depression. As these groups of people are the major potential workforce among the population, the government and society need to find suitable strategies to decrease their depression.

Depressed personnel exhibit emotional disorder syndrome, a mental state which combines several types of emotions, for example, worries, anxiety, sadness and depression [4]. As for treatment, Martinsen [5] suggested that exercise, medication and psychological therapies may decrease depression levels effectively. Although psychological therapies and medication may be necessary for certain serious depression cases, exercise is more flexible and easier to implement into their daily lives. In addition, exercise has no drug side effects. Sport participation also has multiple physiological and psychological benefits. For example, exercises can promote cardiovascular functions, increase bone mass and strengthen immune function, etc. [6]. As for psychological aspects, the exercise engagement process helps the brain to secrete endorphins and dopamine; endorphins and dopamine can ease pains and generate pleasant and positive emotions [7,8,9].

Previous studies have shown the benefits of exercises to health. For example, Chen and Lin [10] investigated 1600 college students’ depression levels using sport participation level and whether they were engaged in sport clubs as predicting variables. The study found students with sport club memberships and higher engagement in exercise had lower levels of depression.

When undergoing exercise, people may have flow experiences. Flow is a state of focusing on the things they are doing and under that state, people enjoy the pleasure fully from doing it and almost forget the time flies. Flow only occurs when people are doing something which is competent and challenging to them [11]. Under such a state, people feel joyful and exhibit a positive mental state [12]. This positive and mindful state helps people to decrease their depression level [13].

As for perceived health status, it is an objective index which can be used as a reference to access a subject’s overall health status and which was proved to be cost-effective compared to physical checkup [14], hence it can also be used as an index of mental status.

Considering the above-mentioned depression-related variables, the study intended to explore the predicting power of sport participation level, flow experience and perceived health status on depression among college students using sex and grades as control variables. The findings may contribute to promoting college students’ engagements in sports and provide insightful suggestions for students and college administrators.

The following paragraph will discuss the previous findings related to the relationships among the predicting variables (sport participation level, flow experience, perceived health status and control variables) and dependent variables (perceived health status and depression).

### 1.1. The Relationship between Sport Participation Level and Perceived Health Status

Sport participation level includes frequency, duration and intensity [15]. As for perceived health status, this index comprises physiological, psychological and social interactions of oneself which covers the personal overall health status as related to one’s physical activity, vigor, sleep, pain, social isolation and emotional reactions [16]. As for sport participation level and perceived health status, Lee [17] found people who regularly engaged in exercise had a higher level of perceived health status and had fewer medical costs. Exercise frequency is found to correlate with perceived health status. For example, studies found on those with an exercise frequency of three or above times per week had a higher perceived health status than those who exercised less [18]. Chang et al. [19] found the positive relationship between sport participation level was positively correlated with perceived health status. Liang, Chen and Huang [20] also found the positive relationship between sport participation level and perceived health status among students. Based on these related studies, the following hypothesis (H1) was proposed:

**Hypothesis** **1 (H1).**
*Sport participation had positive effects on perceived health status.*


### 1.2. The Relationship between Sport Participation Level and Depression

Exercise helps sustain physical fitness and secretes endorphins, dopamine, serotonin and norepinephrine, which can help induce positive emotions, increase concentration, long-term memory and decrease life stress [21]. Serotonin is a type of neurotransmitter which affects the transfer of the impulse to another nerve fiber, a muscle fiber, or some other structure [22]. The serotonin level in blood is highly related to the cause of depression. Moderate physical activity can activate serotonin secretion and decrease depression. Kita [23] also pointed out that exercises can activate serotonin secretion and ease depression levels. Wolf et al. [24] employed a system review targeting physical activity (PA) associations with depression and anxiety during the COVID-19 pandemic. Among the 21 observational studies, they concluded that those performing PA during COVID-19 associated with less depression. Previous studies also found exercise helped ease depression tendencies for college students who regularly engaged in exercise [13,25]. In summary, they concluded that sport-involved adolescents’ depression was significantly lower than in those who were not involved in sport. Panza et al. [26] employed a systematic review and meta-analysis related to 29 adolescents’ organized sport participation and self-reported symptoms of depression. Based on these related studies, the following hypothesis (H2) was proposed:

**Hypothesis** **2 (H2).**
*Sport participation level had negative effects on depression level.*


### 1.3. The Relationship between Perceived Health Status and Depression

Roberts, Dunkle and Haug [27] pointed out that people with higher perceived health status had better behavioral competence and well-being; on the other hand, people with low SRH exhibited negative emotions which also affected their well-being. In other words, physiological status may affect psychological status. As a matter of fact, some studies had found people with lower sport participation levels exhibited higher levels of depression [28,29]. Previous studies also showed sport participation level was a predictor of depression tendency [20,30]. Based on these related studies, the following hypothesis (H3) was proposed:

**Hypothesis** **3 (H3).**
*Perceived health status had a predicting power on depression level.*


### 1.4. Flow Experience

The concept of flow experience was proposed by Dr. Mihaly Csikszentmihalyi [11,31], he described that “flow is a state in which people are so involved in an activity that nothing else seems to matter; the experience is so enjoyable that people will continue to do it even at great cost, for the sheer sake of doing it”. As for a sport participant, flow experience occurs when a sport participant focuses on the activity and is competent in the sport’s skill and feels the utmost pleasure doing it [11,32]. In positive psychology, flow experience is a positive psychological resource, which can generate positive mental toughness [33]. Regarding the influence of flow experience on sport participation motivation, it increased the next-time participation motivation of marathon runners [34], increased jogger sport performances and well-being [35], and according to Tseng, Pi and Lin [36], well-being positively affected perceived health status; hence, flow experience may increase people’s perceived health status and increase recreational sports participants’ life satisfaction [13], etc. In a study focused on elderly people, the level of recreational sports participation affected flow experience directly and affected depression level indirectly. It indicates that recreational sport participation level can increase flow experience and the flow experience can decrease depression [13]. Nien [37] explored various types of sport athletes’ anxiety related to flow experience. The study found the higher the anxiety, the lower the flow experience. Based on these related studies, the following hypotheses (H4 and H5) were proposed:

**Hypothesis** **4 (H4).**
*College students’ flow experience moderates the influence of sport participation level on perceived health status.*


**Hypothesis** **5 (H5).**
*College students’ flow experience moderates the influence of sport participation level on depression.*


Figure 1 illustrates the constructed research hypotheses and we also added two background variables as control variables as predicting variables that may have impacts on perceived health status and depression (H6 and H7) as follows.

**Hypothesis** **6-1 (H6-1).**
*The gender of college students had significant impacts on perceived health status.*


**Hypothesis** **6-2 (H6-2).**
*The gender of college students had significant impacts on depression.*


**Hypothesis** **7-1 (H7-1).**
*The grade of college students had significant impacts on perceived health status.*


**Hypothesis** **7-2 (H7-2).**
*The gender of college students had significant impacts on depression levels.*


## 2. Methods

### 2.1. Participants

A total of 700 questionnaires were distributed to seven universities’ students in Taiwan, each with 100 copies using convenient sampling. All participants consented to the survey and all signed the consent form. Out of them, 605 responses were retrieved with a valid return rate of 86.5%. Table 1 presents the descriptive statistics of the participants. As for gender, there were 320 (52.9%) male, 285 (47.1%) female; as for grade, there were 149 (24.6%) freshman, 155 (25.6%) sophomore, 141 (23.3%) junior and 160 (26.4%) senior.

### 2.2. Measurement

A structured questionnaire consisting of demographical scale, sport participation level scale, flow experience scale, depression level scale, and self-rated health scale was adopted for data collection. The following introduces the origins of the scales.

#### 2.2.1. Demographical Scale

The demographic information includes gender and grade of the participants.

#### 2.2.2. Sport Participation Level Scale

The sport participation level scale was adopted from Fox [38], which includes three items: frequency (how many times you do sports per week?; the frequency scale was rated on a five-point Likert-type scale ranging from 1 (“once”) to 5 (“5 times or more”)), duration (on average, how long do you usually do the sport each time?; the duration scale was rated on a five-point Likert-type scale ranging from 1 (“less than 30 min”) to 5 (“more than 120 min”)) and intensity (light, moderate or heavy; the intensity scale was rated on a five-point Likert-type scale ranging from 1 (“light”) to 5 (“heavy”)). The sport participation level can be estimated using the following formula: sport participation level = frequency × (duration + intensity).

#### 2.2.3. Flow Experience Scale

The flow experience scale was adapted from the flow experience scale developed by Han [39]. The scale consists of five items and was rated by Likert’s 5-point scale ranging from 1 (strongly disagree) to 5 (strongly agree). The higher the score, the higher the flow experience. The Cronbach’ α of the scale is 0.82 indicating high internal reliability.

#### 2.2.4. Depression Level Scale

The depression level scale used in the survey was adopted from Lee et al. [40]. The scale consists of 18 items, the question asks how often does the participant have the feelings per week, the rating scale was categorized using the following: “none or rarely” is scored as 1; “1–2 days per week” is scored as 2; “3–4 days per week” is scored as 3; “5–7 days per week” is scored as 4. The Cronbach’ α of the scale is 0.90 indicating high internal reliability. Clincally, the total score of the 18 items of the participants can be used to categorize the depression level. Under 19: no depression syndrome; 19–23: mild depression; 24–29: middle level of depression; above 30: high level of depression.

#### 2.2.5. Perceived Health Status Scale

The perceived health status scale was adopted from Hunt et al. [16], the scale consists of 6 items and was rated by Likert’s 5-point scale ranging from 1 (strongly disagree) to 5 (strongly agree). The Cronbach’ α of the scale is 0.82 indicating high internal reliability.

### 2.3. Statistical Analysis

In the study, partial least squares structural equation modeling (PLS-SEM) was employed to test the proposed hypotheses [41,42]. According to Pirouz [43], PLS techniques have many advantages compared to traditional covariance-based SEM; for example, PLS can process data without restrictions of data distribution (distribution-free), can process data requiring only a small sample size, has the ability to process multiple dependent and independent variables simultaneously, is able to handle collinearity and is able to process both formative or reflective indicators. With those advantages, PLS-SEM was used to test the proposed hypotheses in the study.

## 3. Results

### 3.1. Descriptive Statistics

Table 2 presents the descriptive statistics of each construct. Most respondents’ sport participation level was classified as low participation level (according to Fox [35], the values indicating sport participation level are: below 15 is low, between 15–45 is medium, and above 45 is high). It indicated college students’ sport engagement is somewhat less than medium and more motivation is still needed for participation. Since the survey was conducted during the pandemic, many sports fields were shut down. Hence students’ sport participation level was low. The mean of flow experience is 3.68, which indicated students experienced some degree of flow during exercises. The mean of perceived health status is 3.67, which indicated students felt their health status was normal-to-good but definitely not excellent. The mean depression level is 1.67, which indicated students perceived low levels of depression, which was a good sign.

### 3.2. Hypothesis Test Results

The hypotheses of this study were tested using Warp PLS 8.0 developed by Kock [44]. PLS analysis comprised two parts: the first part is confirmatory factor analysis, also known as the measurement model, which is performed to examine the reliability and validity of each construct (in our study, the constructs are the aforementioned predicting and dependent variables); the second part is path analysis, also know as the structural model, which calculates the standardized path coefficients of each predicting variables to dependent variables and also tests the significance of each path coefficient. The following sections will present the test results from PLS analysis.

#### 3.2.1. Measurement Model

The reliability and validity of the study instrument were tested using WarpPLS 8.0 [44], which under PLS, provides two measures of item reliability: composite reliability and Cronbach’s α. The convergent validity and discriminant validity were conducted to test the validity of the instrument according to Fornell and Larcker [45]. The factor loading of all items from the PLS measurement model are all greater than 0.70, indicating good indicators. Composite reliability and Cronbach’s α values for all scales exceeded the minimum threshold level of 0.70 (Fornell and Larcker, 1981) indicating the reliability of all scales used in the study (Table 2). As for convergent validity, the square root of the average variation extract (AVE) of all values exceeded the minimum threshold level of 0.70 (Fornell and Larcker, 1981), indicating the reliability of all scales used in the study (Table 2). Fornell and Larcker’s test (1981) for discriminant validity revealed relatively high variances extracted for each factor compared to the interscale correlations, which was an indicator of the discriminant validity of the four constructs (Table 2).

#### 3.2.2. Structural Model

The hypotheses of this study were tested using Warp PLS 8.0 developed by Kock [38]. In the proposed model, the path coefficients represent the positive or negative relationship between variables. If the relationship between two variables exists, the test for the corresponding path coefficient should be significant, and the sign of the path coefficient should also be consistent with the hypothesized direction. With that in mind, the test results are illustrated in Figure 2 and explained in the following paragraph.

H1: Sport participation level had positive predicting power on perceived health status. The path coefficient is 0.42, *p* < 0.05, which indicated that the higher the sport participation level, the higher the perceived health status.

H2: Sport participation level had negative impacts on depression level. The path coefficient is −0.13, *p* < 0.05, which indicated that the higher the sport participation level, the lower the depression level.

H3: Perceived health status had negative predicting power on depression level. The path coefficient is −0.69, *p* < 0.05, which indicated that the higher the perceived health status, the lower the depression level.

H4: Flow experience had no moderating effects on sport participation level and perceived health status. The path coefficient to perceived health status is −0.02, *p* > 0.05, which indicated no moderation effects among them.

H5: Flow experience had no moderating effects on sport participation level and depression level. The path coefficient to depression level is −0.02, *p* > 0.05, which indicated no moderation effects among them.

H6-1: The gender of college students had negative impacts on perceived health status. The path coefficient is −0.07, *p* < 0.05, which indicated male students had higher perceived health status than female students.

H6-2: The gender of college students had no significant impact on depression. The path coefficient is 0.06, *p* > 0.05, which indicated there were no differences in depression levels between male and female students.

H7-1: The grade of college students had no significant impacts on perceived health status. The path coefficient is 0.06, *p* > 0.05, which indicated there were no differences of perceived health status perceptions among all grade levels of college students.

H7-2: The gender of college students had negative impacts on depression levels. The path coefficient is 0.07, *p* < 0.05, which indicated higher-grade college students had higher depression level perceptions.

### 3.3. Explanatory Power (R^2^)

R^2^ indicates the predicting power of predicting variables to the outcome variables (which in this case, were perceived health status and depression). It refers to the predictive power of the research model [46]. Cohen [47] used R^2^ to explain the effect size of the model performance, i.e., R^2^ < 0.10 indicates a small effect size to the outcome variable; 0.10–0.30 indicates a medium effect size; R^2^ > 0.50 indicates a large effect size. In the study, the effect size for perceived health status was medium and the effect size for depression was large, indicating a predicting power.

## 4. Discussion

### 4.1. The Relationship of Gender to Perceived Health Status and Depression

The study found there was no significant difference in depression between male and female college students. However, male students’ perceived health status perception was higher than female students, which was consistent with previous findings [34,37,39]. Tseng, Pi and Lin [36] also found that most sports centers’ members were male and aware of body fitness and willing to work out to maintain attractiveness. Interestingly, according to Jackson and Marsh [45], it is a stereotype for women. Traditionally, people believe manly guys engage in all kinds of sports and through these, they can gain positive recognition from society; on the other hand, female sports lovers may be viewed as masculine. In fact, the ratio of regular exercise participants of males and females exhibited a significant difference. According to the Sport Administration, Ministry of Education, Taiwan (SAMET) [46], the ratio of regular exercise participants among 13–17 years groups was 53.5% for males, and 28.9% for females; among 18–24 groups was 35.9% for male and 27.9% for female. The statistics all indicated men are more attached to exercise than women. Therefore, the findings from our study also showed that male students had higher perceived health status perception than females.

### 4.2. The Relationship of Grade to Perceived Health Status and Depression

The study found there was no significant difference in perceived health status perception among all grade groups. However, college students with higher grades had higher depression than students with lower grades. The findings were consistent with previous studies [48]. Compared to senior and junior students, freshmen and sophomores had no urgent necessities to look for jobs and hence were less stressed [49,50,51]. Huang and Lin [52] also found junior and senior college students exhibited higher anxiety toward the future than younger students. Compared to students with lower grades, senior students need to prepare for work and may consider getting married, hence they may undergo more pressure than younger students.

### 4.3. The Relationship between Sport Participation Level and Perceived Health Status

The study found college students’ sport participation levels had positive predicting power on perceived health status, which is consistent with previous research findings [49,50]. Lin and Lin [51] found those who engaged in sport regularly perceived higher perceived health status than those who seldom participate in sports. Chen, Lin and Pi [52] investigated street dance participants; the results showed that most of them recognized the benefits of engagement in street dance, such as improvement of cardiovascular function, better motor skills, improvement of physical ability and fitness, making new friends and less loneliness. Since perceived health status is highly correlated with objective health, college students are advised to become active and engage in sports which can help them be healthier. According to the Sports Administration, Ministry of Education, Taiwan [53], in 2013, an investigation of 8,381 college students’ sport participation constraints showed the top three factors were: lack of time (44.3%), no suitable sports field (919.9%) and no company (11.1%). It is, therefore, suggested that college students should better manage their schedule to make time for exercise to improve their health.

### 4.4. The Relationship between Sport Participation Level and Depression

The study found college students’ sport participation levels had negative impacts on depression, which was consistent with some previous findings [54,55]. According to Chen and Lin [10], college students who participated in sports regularly were less depressed than those who rarely did. Martines-Gonzalez [56] also pointed out that people who engaged in sports more often can be healthier than others and enhance their wellbeing. Huang, Wang and Lu [57] pointed out that higher sport participation level personnel can gain happiness, release pressure, promote physiological health and decrease anxiety. It is, therefore, suggested that college students stick to exercise.

### 4.5. The Relationship between Perceived Health Status and Depression

The study found that perceived health status had a negative predicting power on depression. This finding supports some previous related findings [20,28,29,30] which state that perceived health status was negatively corrected with depression. Lin et al. [58] pointed out depression was negatively correlated to perceived health status; Liang, Chen and Chen [20] investigated 1194 college students located in northern Taiwan using random sampling. The study also found students with higher perceived health status had lower depression. Liu [29] also concluded that physiological status was correlated with psychological status. Therefore, personnel with poor perceived health status perception might encounter bad emotions, which could cause depression.

### 4.6. The Relationship of Flow Experience among Sport Participation Level and Depression

The study found no significant moderating effects among sport engagement and depression; it is not consistent with Chang’s findings [13]. According to Csikszentmihalyi [59], flow consists of three basic elements: concentration, interest and enjoyment. In other words, flow experience occurs when personnel concentrate on his/her interested activities and fully focus and enjoy the moment. While the study did not find significant moderating effects among sport engagement and depression, we found several possible causes. For one, the investigation was conducted during the SARS-CoV-2 pandemic. Due to the pandemic, students were enforced to wear masks all the time and most sports centers were shut down, which included swimming pools, gyms, parks and some sports fields. In addition, physical education classes were transformed into online courses. During the quarantine periods, students could not choose their favorite sports field for exercises. These factors might affect their flow experience.

## 5. Conclusions and Suggestions

### 5.1. Conclusions

This study examined the relationships of college students’ sport participation level, flow experience, perceived health status and depression using gender and grade as control variables. The study found male students had higher perceived health status than female students and students with higher grades exhibited higher depression than students with lower grades. As for sport participation level, it positively affected perceived health status and negatively affected depression; perceived health status had negative predicting power on depression; flow experience had no moderating (enhancing) effects among the relationships of sport participation level to perceived health status and depression. The hypothesized model was able to explain 58% variance of college students’ depression indicating a well-constructed model.

### 5.2. Suggestions

According to research findings, the study provides practical suggestions for college students and university administrators.

First, for college students: according to Sparling and Snow [60], their longitudinal survey on college students’ exercise behaviors found the best timing of cultivating exercise habits and behaviors was during college life. Their statistics showed that among investigated students who participate in sports regularly, 84.7% of them maintained their exercise routine after graduation. On the other hand, 81.3% students with no regular exercise behaviors remained inactive after graduation. Therefore, we strongly encourage college students to put exercise on calendars and make exercise their daily routine. In that way, their perceived health status can be elevated, they can obtain self-confidence, decrease depression and become happier. 

The study also found female students’ perceived health status was lower than that of male students. Therefore, it is suggested especially for female students to join a sports club and plan to do exercise during their leisure time.

Second, for university administrators: even though physical education courses are required courses for freshmen and sophomores in Taiwan, most college students had little intention to elect physical education courses after completing the required credits. The reason may be that there are few varieties of professional physical education courses for students to choose from. Therefore, the study recommends universities provide varieties of physical education courses and hold more sports events to cultivate a sport-friendly environment for students to increase their exercise motivation. In addition, the physical education courses were usually practiced in a large crowd. In each of these courses, there were usually 40–70 students in a class with students in different motor skill levels. Therefore, we strongly recommend the university physical education office implements a motor skill test before class and assigns students with similar motor skill levels into smaller groups. In that way, teachers can focus on helping students increase motor skills, cultivate interests in sports and enjoy flow experience, which can definitely help them in building up exercise habits in their daily lives.

The survey of this study was conducted during the pandemic under which most of the sports fields were shut down and physical education classes were transformed into online courses. Those situations might affect students’ sports participation motivation and flow experience. Therefore, the study suggests performing the survey post-pandemic and re-examining the hypotheses in the study. The cross-examination can also compare students; engagements in sports and flow experience during and after the pandemic. Another suggestion is to use a hierarchical linear method to test the proposed hypotheses which can identify the school’s effects on students since different universities exhibit different cultures, and by using advanced analytical techniques, we can get clearer insights and make more practical implications.

## Figures and Tables

**Figure 1 ijerph-20-00251-f001:**
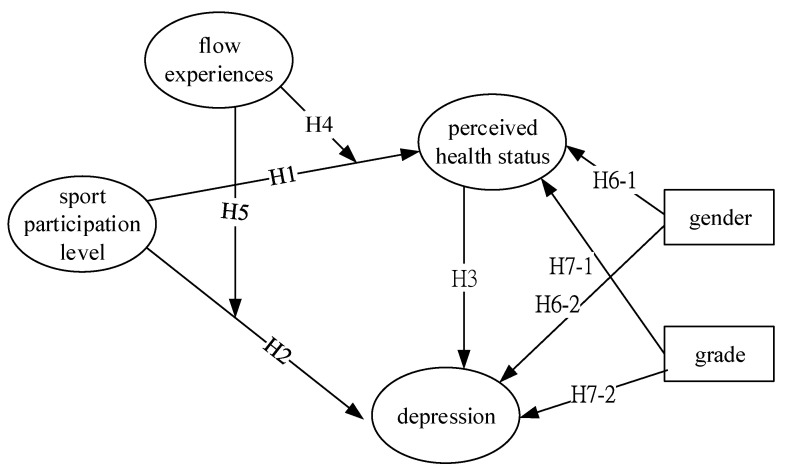
Research hypotheses framework.

**Figure 2 ijerph-20-00251-f002:**
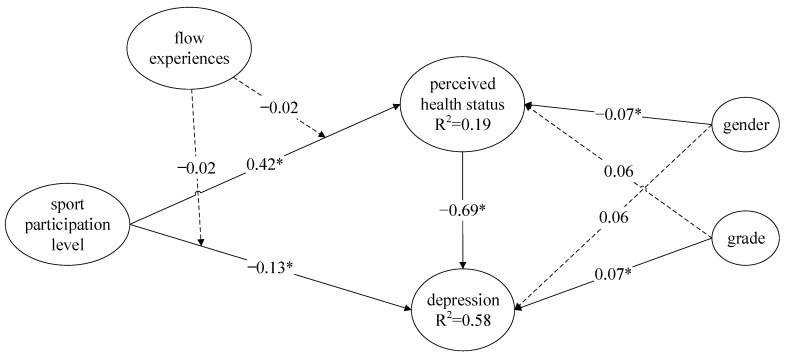
SEM results of the standardized model parameter estimation. Note: “...” represents “path coefficient was not significant” “–” represents “path coefficient was significant”. ** p* < 0.05.

**Table 1 ijerph-20-00251-t001:** Descriptive statistics of participants (N = 605).

Variable	Group	n	%
Gender	Male	320	52.9
	Female	285	47.1
Grade	Freshman	149	24.6
	Sophomore	155	25.6
	Junior	141	23.3
	Senior	160	26.4

**Table 2 ijerph-20-00251-t002:** Reliability, convergent and discriminant validity of measurement model.

Construct	M	SD	(1)	(2)	(3)	(4)	CR ^b^	A ^c^
(1) Sport participation level	12.14	6.86	1.00 ^a^				1.00	1.00
(2) Flow experience	3.68	0.65	0.38	0.76 ^a^			0.87	0.82
(3) Perceived health status	3.67	0.81	0.45	0.26	0.73 ^a^		0.87	0.83
(4) Depression level	1.63	0.54	−0.34	−0.27	−0.72	0.70 ^a^	0.94	0.94

Note: M: mean; SD: standard deviation; ^a^: square root of AVE (average variance extracted); ^b^: composite reliability; ^c^: Cronbach’s α.

## Data Availability

Data can be obtained upon request.

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
