# Peer review of "The Relationships among Sport Participation Level, Flow Experience, Perceived Health Status and Depression Level of College Students"

_ijerph, 2022, doi:10.3390/ijerph20010251_

Round 1

Reviewer 1 Report

Reading the submitted article was a pleasure. The literature review developed contained all the necessary aspects of the topic covered. The hypotheses presented succinctly form a logical system of links that has been methodologically sound. The only caveat that can be made is the age of the literature used as only about 27% of the items are from the last 5 years (2018 and newer). 

Author Response

Comments and Suggestions for Authors

Reading the submitted article was a pleasure. The literature review developed contained all the necessary aspects of the topic covered. The hypotheses presented succinctly form a logical system of links that has been methodologically sound. The only caveat that can be made is the age of the literature used as only about 27% of the items are from the last 5 years (2018 and newer). 

Response: Thank you very much for reviewing this article and your suggestions. Your suggestions are highly appreciated. We had added two recent articles in the literature review section. Please refer to section 1.2. The articles are:

Panza, M. J., Graupensperger, S., Agans, J. P., Doré, I., Vella, S. A., & Evans, M. B. (2020). Adolescent sport participation and symptoms of anxiety and depression: A systematic review and meta-analysis. Journal of sport and exercise psychology42(3), 201-218. Doi: 10.1123/jsep.2019-0235

Wolf, S., Seiffer, B., Zeibig, J. M., Welkerling, J., Brokmeier, L., Atrott, B., ... & Schuch, F. B. (2021). Is physical activity associated with less depression and anxiety during the COVID-19 pandemic? A rapid systematic review. Sports Medicine51(8), 1771-1783. Doi:10.1007/s40279-021-01468-z

Thank you.

Reviewer 2 Report

In general I find the topic of the article very interesting and very relevant that the relationships between sport participation level, perceived health status and depression level of college students are explored. I consider the flow variable less relevant and, in fact, it has not offered interesting results. I like very much the presentation of the hypotheses and the relationship between the variables studied in graphical form, but as far as the results are concerned I miss at least two tables of results. A table with the descriptive values and reliability indicators of the instruments used for the group of participants and a more detailed representation of the contrast values, which would allow the inclusion of this study in subsequent meta-analyses, or the use of the results for a detailed comparison in the discussion of future studies.
Another element that seems essential to me is the code of reference to the report of the bioethics committee of the university or the body responsible for the research. I consider insufficient the statement that All participants consented to the survey and all signed the consent form.
Also a procedural section detailing how the sample selection was developed, what format was chosen for the presentation of the evidence, etc.
Finally, in the suggestions section I note a certain insistence on the part of the researchers on the need to encourage flow experiences among the participants, when their results do not provide any evidence in this regard.

Author Response

Comments and Suggestions for Authors

 In general I find the topic of the article very interesting and very relevant that the relationships between sport participation level, perceived health status and depression level of college students are explored. I consider the flow variable less relevant and, in fact, it has not offered interesting results. I like very much the presentation of the hypotheses and the relationship between the variables studied in graphical form, but as far as the results are concerned I miss at least two tables of results. A table with the descriptive values and reliability indicators of the instruments used for the group of participants and a more detailed representation of the contrast values, which would allow the inclusion of this study in subsequent meta-analyses, or the use of the results for a detailed comparison in the discussion of future studies.
Another element that seems essential to me is the code of reference to the report of the bioethics committee of the university or the body responsible for the research. I consider insufficient the statement that All participants consented to the survey and all signed the consent form.
Also a procedural section detailing how the sample selection was developed, what format was chosen for the presentation of the evidence, etc.

Response: Thank you for your comments. We had added Table 2 to show the descriptive statistics of all variables and the results of validity and reliability test. Please refer to sections 3.1 and 3.2 and also Table 2. As for the ethics of the survey, well, in fact, we have many associates among these colleges. Well, to be clearer, students are more willingly to sign the consent form than western countries’ students due to the influence of Confucius philosophy. When their instructors invited them to participate in the survey, most of them agreed to sign the consent form. That was what happened in our survey. The consent is provided in the following for your reference:

Consent form

This voluntary survey is part of a research study led by the Tamkang University. The purpose of this research is to gain a better understanding of college students’ sport participation level, flow experience, perceived health status and depression level. This survey will take about 10 minutes each time.

In order to ensure that we cannot identify you and to keep your responses confidential, we do not collect your name or any other personal data from you.

Published results will be aggregated and will not identify you individually or your responses.

If you have questions about the study please contact:

[email protected]

You understand the above and consent to take part in this survey run by the Tamkang University.

â–¡Yes

â–¡No

Finally, in the suggestions section I note a certain insistence on the part of the researchers on the need to encourage flow experiences among the participants, when their results do not provide any evidence in this regard.

Response: Thank you for pointing that out. We had deleted such suggestions.

Reviewer 3 Report

Thank you for the opportunity to review this article. This manuscript is well written and well informed. However, I have some concerns about its contribution which means that it is not currently suitable for publication.

Main concern.

The moderating effect of flow experiences was not significant. While the investigation of the association of the three variables (sport participation, perceive health status, and depression) lacks innovation. So what’s the contribution of the present study?

Some other comments for the authors’ reference.

The study established a mediating model (Figure 1). However, the rational of the mediation of perceived health status is lacked. I would suggest the authors supplement why perceived health status play a mediating role between sport participation level and depression in the introduction.

Statistical analysis in the method part is lacked.

The present study is cross-sectional in nature, so directional conclusions could not be drawn. I would suggest the authors revise the terminologies such as “predicting” “effects on” “affected”.

L174 check for the formula. sport participation level = frequency × intensity × (duration -1)?

L1 “pf” Check for errors.

L314 Check for errors.

Author Response

Comments and Suggestions for Authors

Thank you for the opportunity to review this article. This manuscript is well written and well informed. However, I have some concerns about its contribution which means that it is not currently suitable for publication. 

Main concern.

The moderating effect of flow experiences was not significant. While the investigation of the association of the three variables (sport participation, perceive health status, and depression) lacks innovation. So what’s the contribution of the present study? 

Response: Thank you for your comments. Well, in fact, many factors affect depression. This study wanted to explore the relationships among those variables. While we didn’t find flow experience was a moderating variable among perceived health status and sport participation level to depression level, we did find perceived health status and sport participation level predicted depression level. Even so, we did discuss the possible reasons and compared with some previous studies. In addition, we also made suggestions according to our findings and we believe those findings and suggestions contribute to advises for both college students and university administrators in the promotion of sport participation.

Some other comments for the authors’ reference.

The study established a moderating model (Figure 1). However, the rational of the moderating of perceived health status is lacked. I would suggest the authors supplement why perceived health status play a moderating role between sport participation level and depression in the introduction.

Response: Thank you for the suggestions. We had explained the connections among flow experience, perceive health status, and depression. Please check section 1.4. Thank you.

Statistical analysis in the method part is lacked.

Response: We had added the statistical analysis section. Please refer to section 2.3 Statistical analysis. Thank you.

 The present study is cross-sectional in nature, so directional conclusions could not be drawn. I would suggest the authors revise the terminologies such as “predicting” “effects on” “affected”.

Response: Thank you for your comments. The terminology used in this article was based on the proposed hypotheses and the underlying reason we didn’t use “affect” term is that “perceived health status” is actually a type of self- perception, not a influencing factor and can only be used as a indicator or a predictor to evaluate depression level. Therefore, we feel using perceived health status as a predictor to depression level is appropriate in our case.    However, it may be confusing to readers. Therefore, we revised some terms in the final part of our article. Especially in the discussion sections. Thank you. 

L174 check for the formula. sport participation level = frequency × intensity × (duration -1)?

Response: Thank you for your comments. We checked the formula again and it is correct. The formula was first proposed by Fox (1999).

Fox, K. R. The influence of physical activity on mental well-being. Public Health Nutrition, 1999, 2(3), 411-418.

L1 “pf” Check for errors.

Response: Thank you for the reminders. We corrected this mistake. Thank you.

L314 Check for errors.

Response: Thank you for the reminders. We had corrected this error.

Round 2

Reviewer 2 Report

From the very first reading, I found the article very interesting. The authors have accepted all my comments and suggestions, completing some incomplete aspects of the methodology and bringing greater coherence to the discussion of the results. There is only one important aspect that they have not been able to include, and that is the reference to the approval of the study by a bioethics committee linked to the institution to which they belong. However, they do provide the consent form filled in by the participants in the study. From my point of view, the work deserves to be published, but I understand that the ethical aspects are relevant and I leave this decision to the discretion of the journal's editors.

Author Response

Reviewer 2

comments and suggestions, completing some incomplete aspects of the methodology and bringing greater coherence to the discussion of the results. There is only one important aspect that they have not been able to include, and that is the reference to the approval of the study by a bioethics committee linked to the institution to which they belong. However, they do provide the consent form filled in by the participants in the study. From my point of view, the work deserves to be published, but I understand that the ethical aspects are relevant and I leave this decision to the discretion of the journal's editors.

Response: Thank you for your comments.

Reviewer 3 Report

Thank you for the response. I have one more comment about the formula. It seems ridiculous to add up the two (duration + intensity) as they have different units. I would suggest the authors check the reference again. 

Author Response

Reviewer 3

add up the two (duration + intensity) as they have different units. I would suggest the authors check the reference again. 

Response: Thank you for your comments. The formula is checked out. We also changed the citing article as: Fox, K. R. Physical self-perceptions and exercise involvement. [Doctoral dissertation]. ProQuest Dissertations Publishing. Arizona State University,1987.

To be clearer, we revised the description of this formula as the following:

2.2.2. Sport participation level scale

The sport participation level scale was adopted from Fox [38], which includes three items: frequency (how many times you do sports per week?; The frequency scale was rated on a five-point Likert-type scale ranging from 1 (‘‘once’’) to 5 (‘‘5 times or more’’).), duration (in average, how long do you usually do the sport each time?; The duration scale was rated on a five-point Likert-type scale ranging from 1 (‘‘less than 30 minutes’’) to 5 (‘‘more than120 minutes’’). ) and intensity (light, moderate, or heavy; The intensity scale was rated on a five-point Likert-type scale ranging from 1 (“light”) to 5 (“heavy” ). The sport participation level can be estimated using the following formula: sport participation level = frequency ï‚´ (duration+ intensity).

Thank you very much.
